# Bovine Milk Fat Globule Membrane Supplementation and Neurocognitive Development: A Systematic Review and Meta-Analysis

**DOI:** 10.3390/nu16142374

**Published:** 2024-07-22

**Authors:** Therdpong Thongseiratch, Kulnipa Kittisakmontri, Nutthaporn Chandeying

**Affiliations:** 1Child Development Unit, Department of Pediatrics, Faculty of Medicine, Prince of Songkla University, Songkhla 90100, Thailand; 2Division of Nutrition, Department of Pediatrics, Chiang Mai University, Chiang Mai 50200, Thailand; 3Department of Obstetrics and Gynecology, Faculty of Medicine Vajira Hospital, Navamindradhiraj University, Bangkok 10300, Thailand; nutthaporn027@gmail.com

**Keywords:** bovine milk fat globule membrane, neurocognitive development, infant formula, cognitive development, infant nutrition

## Abstract

Given the limited evidence, there is no conclusive proof of the neurocognitive benefits of bovine milk fat globule membrane supplementation in infant formula. This study evaluates the neurocognitive benefits of bovine milk fat globule membrane supplementation in formula, comparing it to standard formula and assessing its noninferiority to breast milk. Data were sourced from studies published between January 2000 and March 2024 from PubMed, Cochrane Library, Web of Science, and Embase. Eight randomized controlled trials involving 1352 healthy term neonates, infants, and children up to 2 years old were included. Bovine milk fat globule membrane supplementation was significantly associated with improved cognitive development (mean difference: 3.29, 95% CI: 1.65 to 4.93, *p* < 0.001) and demonstrated minimal heterogeneity (*I*^2^ = 0%, *p* = 0.564). It showed significant improvement in executive function but not in language, motor, or social-emotional development. In non-inferiority analysis, there was no significant difference compared to breast milk regarding cognitive development. These findings support bovine milk fat globule membrane as a valuable addition to infant formula for cognitive benefits.

## 1. Introduction

Breast milk is widely recognized as the gold standard for infant nutrition, providing optimal nutrients, immune factors, and bioactive molecules essential for healthy development [1]. Given its unmatched benefits, infant formula manufacturers continuously strive to replicate these advantages to achieve noninferiority in terms of nutritional and developmental outcomes [2,3]. This ongoing research has led to numerous formulation adjustments aimed at closing the gap between breast milk and formula [4]. The ultimate goal is to create a product that can offer similar health benefits to those who are unable to breastfeed, thereby supporting the healthy growth and development of all infants.

One notable advancement in infant formula is the inclusion of supplements such as long-chain polyunsaturated fatty acids (LCPUFAs), which support brain and visual development [5,6,7]. LCPUFAs are critical for the development of the nervous system and visual acuity, playing a key role in the structural composition of cell membranes in the brain and retina. Additionally, prebiotics are added to enhance gut health by promoting the growth of beneficial gut bacteria, which can improve digestion and strengthen the immune system. Furthermore, choline, taurine, and lutein are incorporated to promote normal neural and visual development [8]. Choline is vital for brain development and function, taurine supports the development of the central nervous system, and lutein is an antioxidant that contributes to eye health.

Recent research has focused on supplementing formulas with bovine milk fat globule membrane (MFGM), a component naturally present in breast milk that is critical for brain structure and function [9,10,11]. MFGM contains essential elements such as sphingomyelin and gangliosides, which play significant roles in neurodevelopment and cognitive function [11]. Sphingomyelin contributes to myelin sheath stability and neural signaling [12], while gangliosides are involved in cell interactions and brain signaling [13]. Additionally, MFGM includes proteins and cholesterol that support cognitive development and neural function, such as butyrophilin and lactadherin, which are important for brain and immune function [14], and cholesterol, which is vital for myelin sheath integrity and synaptic function [15]. These components collectively enhance the neurodevelopmental outcomes in infants and children, making MFGM an attractive addition to infant formula.

A previous meta-analysis on MFGM supplementation in children, which encompassed 24 publications from 17 studies, primarily focused on growth parameters and provided a narrative review of neurocognitive outcomes. The findings suggested promising effects, including a lower incidence of acute otitis media and some cognitive improvements, alongside a good safety profile for MFGM-supplemented formulas. However, due to limited evidence, the study did not conclusively demonstrate significant neurocognitive benefits, highlighting the need for more well-designed trials to firmly establish these benefits [16]. This underscores the importance of continued research in this area to validate the potential cognitive advantages of MFGM supplementation and to provide clearer guidance for healthcare providers and parents.

In light of these gaps, our study systematically evaluates the neurocognitive benefits of MFGM-supplemented infant formula compared with standard infant formula. This study addresses two primary questions: (1) What is the effect of MFGM on the neurocognitive development of neonates, infants, and children up to 2 years old? This includes assessments of cognitive, language, and motor development, comparing MFGM-supplemented infant formula to standard infant formula. (2) Does MFGM-supplemented infant formula demonstrate noninferiority to breast milk regarding neurocognitive outcomes? This comparison is crucial for validating MFGM-supplemented infant formula as a viable alternative when breastfeeding is not possible or preferred. By addressing these questions, we aim to provide a comprehensive understanding of the potential cognitive benefits of MFGM and its role in infant nutrition, thereby supporting the development of evidence-based recommendations for infant feeding practices.

## 2. Materials and Methods

### 2.1. Study Design

This systematic review and meta-analysis was registered with PROSPERO (ID: CRD42024517716) and is reported following the PRISMA 2020 guidelines [17] and Cochrane Handbook standards [18]. The study aimed to provide a rigorous evaluation of the neurocognitive benefits of bovine milk fat globule membrane (MFGM) supplementation in infant formula compared to standard formulas and breast milk. The study design ensured adherence to systematic review methodology to ensure accurate and reliable results.

### 2.2. Data Sources and Search Strategy

A comprehensive literature search was conducted across PubMed, the Cochrane Library, Web of Science, and Embase. Keywords and MeSH terms related to “infant formula”, “neurocognitive development”, and “MFGM” were used with the assistance of a medical librarian. The search, conducted on 14 April 2024, covered studies published from January 2000 to March 2024. Detailed search strategies for each database included specific terms and Boolean operators to ensure thorough identification of relevant studies. The detailed search strategies for each database ensured the comprehensive identification of relevant studies. For PubMed: (“infant formula”[MeSH Terms] OR “infant formula”[All Fields]) AND (“neurocognitive development”[MeSH Terms] OR “neurocognitive development”[All Fields]) AND (“milk fat globule membrane”[MeSH Terms] OR “milk fat globule membrane”[All Fields]). For Cochrane Library: (“infant formula” OR “neurocognitive development” OR “milk fat globule membrane”) in Title Abstract Keyword. For Web of Science: TS = (“infant formula” AND “neurocognitive development” AND “milk fat globule membrane”). For Embase: (‘infant formula’/exp OR ‘infant formula’) AND (‘neurocognitive development’/exp OR ‘neurocognitive development’) AND (‘milk fat globule membrane’/exp OR ‘milk fat globule membrane’). These strategies ensured all relevant studies were identified for a robust and thorough evaluation.

### 2.3. Eligibility Criteria

The systematic review followed the PICOS framework: Population (P) included neonates and infants aged 0–12 months and children up to 2 years old, excluding those with medical conditions affecting neurocognitive outcomes; Intervention (I) was infant formula supplemented with bovine MFGM provided during the first six months of life; Comparator (C) included standard infant formulas without MFGM and breast milk; Outcomes (O) involved measures of neurocognitive development using validated tools; and Study Design (S) included only randomized controlled trials (RCTs). Exclusions applied to non-English publications and studies lacking detailed formula composition data to maintain high-quality findings. 

### 2.4. Study Selection

T.T. and a research assistant independently screened article titles and abstracts for potential inclusion, followed by a full-text review. Discrepancies were resolved through discussion or consultation with a third reviewer, N.C. This two-tiered process ensured a thorough and unbiased selection of studies, enhancing the reliability of the systematic review by incorporating multiple perspectives and checks.

### 2.5. Data Extraction 

Data were extracted independently by T.T. and N.C. on study characteristics, sample size, location, population demographics, intervention details, comparison groups, and neurocognitive development measures using a standardized form. Following PRISMA guidelines, comprehensive data were collected for all predefined outcomes, ensuring consistency and thoroughness in capturing relevant information across studies, thus, facilitating accurate analysis.

### 2.6. Statistical Analysis

Comprehensive Meta-Analysis software, version 4 (Biostat, Englewood, NJ, USA) [19], was used for the meta-analysis, employing a random-effects model to account for potential heterogeneity assessed by the *I*^2^ statistic. Separate meta-analyses were conducted for each neurocognitive outcome, using the last provided time point for studies with multiple time points. Mean difference was used for measures standardized around 100, and standardized mean difference (SMD) was used for pooling standardized outcomes. The noninferiority margin was set at 50% of the neurocognitive impact of breast milk from a previous meta-analysis, with a predefined margin of 2.66 [20].

### 2.7. Quality Assessment

The quality of included studies was independently assessed by T.T. and a research assistant using the Cochrane Risk of Bias tool for RCTs, covering domains such as selection, performance, detection, attrition, and reporting biases. Visual examinations of funnel plots and Egger’s regression test were used to assess publication bias when enough studies were available. No automation tools were used in this process, ensuring a detailed and manual evaluation of study quality.

## 3. Results

Figure 1 shows the study selection process. Initially, 3458 studies were identified from the electronic databases. After removing duplicates, 531 studies were screened, and 67 full-text articles were evaluated. Following a detailed review and exclusion of non-eligible studies, one additional study was identified through alternative methods, including website searches, Google Scholar searches, citation chasing, and reference lists of existing systematic reviews, resulting in the inclusion of eight studies.

### 3.1. Included Studies

This systematic review included eight studies comparing the neurocognitive development of 1352 healthy, term neonates, infants, and children supplemented with MFGM with that of those who received standard formula or breast milk [21,22,23,24,25,26,27,28]. These studies were conducted in China (n = 4), Spain (n = 2), Indonesia, and Sweden. Interventions involved MFGM-supplemented infant formula, often combined with nutrients such as lactoferrin, LCPUFAs, symbiotics, gangliosides, nucleotides, and sialic acids, starting from as early as the first 2 weeks of life and continuing up to the first year, with follow-up assessments extending to 5.5 years in some cases. Notably, Colombo et al. (2023) [22] is a follow-up study of Li et al. (2019) [24], and the review includes two studies with different neurocognitive outcomes from Nieto-Ruiz et al. (2020) [25,26]. Comparison groups included neonates, infants, and children who received standard formula and those who were breastfed, allowing for a comprehensive evaluation of the association of MFGM supplementation with neurocognitive development. Outcomes were measured at various stages using standardized tools, including the Children Neuropsychological and Behavior Scale-Revision 2016, Wechsler Preschool and Primary Scale of Intelligence—Fourth Edition, Child Behavior Checklist, Stroop Task, Denver Developmental Screening Test, Griffiths Mental Development Scales, and Bayley Scales of Infant Development (Table 1). 

Almost all included studies assessed infant growth as a function of feeding. Chen et al. [21] reported no significant differences in weight among groups at three time points. Colombo et al. [22] did not provide weight data, but Li et al. [24] found similar weights between control and MFGM + lactoferrin groups at birth and enrollment. Gurnida et al. [23] reported no significant differences in birth weight or weight-for-age and weight-for-length z-scores. Nieto-Ruiz et al. [25,26] found no significant differences in BMI at 4 years. Timby et al. [27] reported no significant differences in birth weight and other anthropometric measures at 12 months. Xia et al. [28] found no significant differences in birth weight among groups. The standard formula used in the studies varied by manufacturer and formulation. Each study utilized a commercially available standard formula as the control, which did not contain MFGM supplementation. While the basic composition was similar, the specific ingredients and nutrient profiles differed between products. Additionally, some studies reported dual feeding practices, where infants received both formula and breast milk. This was noted in the studies by Chen et al. [21], Gurnida et al. [23], and Xia et al. [28], where overlapping feeding methods were considered in the analysis.

### 3.2. Meta-Analysis Results: Additional Value to Infant Formula

#### 3.2.1. Cognitive Development 

Our meta-analysis of five RCTs evaluated the association between MFGM supplementation of infant formula and cognitive development. The pooled mean difference was 3.29 (95% confidence interval [CI]: 1.650 to 4.928, *p* < 0.001), indicating that infants who received MFGM-supplemented formula had cognitive scores averaging 3.29 points higher than did those who received standard formula (Figure 2A). This significant difference suggests that MFGM supplementation is associated with enhanced cognitive abilities. The SMD was 0.35 (95% CI: 0.172 to 0.518), reflecting a moderate and statistically significant effect size (Figure 2B). Minimal heterogeneity was observed (Q = 2.96, df = 4, *I*^2^ = 0.00%, *p* = 0.564), indicating consistent positive effects across various populations and settings.

#### 3.2.2. Language, Social–Emotional, Motor, and Executive Function Development

Our meta-analysis evaluated the association between MFGM supplementation of infant formula and various developmental domains (Figure 3). In terms of language development, six RCTs showed a pooled SMD of 0.14 (95% CI: –0.018 to 0.300, *p* = 0.08), indicating no significant improvement with no heterogeneity (*I*^2^ = 0.00%) (Figure 3A). Motor development, assessed in five studies, showed a pooled SMD of 0.29 (95% CI: –0.120 to 0.692, *p* = 0.167), also indicating no significant improvement but with substantial heterogeneity (*I*^2^ = 83.71%) (Figure 3B). Social–emotional development, based on four studies, had an SMD of 0.11 (95% CI: –0.054 to 0.277, *p* = 0.186), with consistent findings (*I*^2^ = 0.00%) (Figure 3C). Executive function, assessed in two studies—one focusing on executive function and the other on attention in toddlers—showed a pooled SMD of 0.48 (95% CI: 0.184 to 0.772, *p* = 0.001), with no heterogeneity (*I*^2^ = 0.00%) (Figure 3D).

### 3.3. Secondary Result: Noninferiority to Breast Milk

#### 3.3.1. Cognitive Development 

Using a random effects model, our meta-analysis assessed the noninferiority of infant formula supplemented with MFGM to breast milk in terms of cognitive development. The point estimate of the effect size was 0.580 (95% CI, -1.391 to 2.551; *p* = 0.564), indicating that the result was not statistically significant (Figure 4). Given the predefined noninferiority margin of 2.66, the upper limit of the CI (2.551) nearly reached but did not exceed this threshold. This outcome suggests that the MFGM-supplemented formula could be considered not inferior to breast milk, as the upper limit of the CI does not surpass the noninferiority margin, implying that any potential superiority of breast milk does not achieve a clinically significant magnitude as per the noninferiority criteria. The heterogeneity among the studies was minimal, with an I^2^ of 0%, indicating no significant variability attributable to heterogeneity across the studies (Table 2). 

#### 3.3.2. Language, Social–Emotional, and Motor Development 

The non-inferiority results in terms of language, social–emotional, and motor development were inconclusive: language development showed a mean difference of 1.91 with a CI of –2.046 to 4.074; motor development, a mean difference of 1.20 with a CI ranging from –0.66 to 3.06; and social–emotional development, a significant mean difference of 3.41 and a wide CI of –4.65 to 11.47. The wide ranges and crossing of zero in these intervals render the noninferiority of MFGM-supplemented formula to breast milk inconclusive in terms of language, motor, and social–emotional development (Table 2). 

### 3.4. Publication Bias

Visual inspection of the funnel plot of the main analysis (Figure 5) showed no apparent signs of publication bias. The distribution of the effect sizes was fairly symmetrical, with most falling within the funnel, indicating a uniform distribution. The few effect sizes falling outside the funnel were symmetrically distributed, suggesting no systematic skewing of the results (Figure 5C). This symmetry supports the reliability and robustness of the meta-analysis across different study sizes and conditions.

### 3.5. Risk of Bias

The risk of bias assessment for the included studies revealed that most had a low risk of bias, particularly in the domains of randomization, deviations from the intended interventions, measurement of the outcomes, and selective reporting (Figure 6) [22,23,24,26,27,28]. However, some studies exhibited concerns, particularly regarding the randomization process, handling of missing outcome data, and blinding of the outcome assessment [21,25]. These findings ensure the robustness and reliability of our meta-analysis despite some variability in individual study quality.

## 4. Discussion

Our systematic review and meta-analysis show that the addition of MFGM to infant formula is significantly associated with an improvement in cognitive development compared to standard infant formula. Neonates, infants, and children who received the MFGM-supplemented formula had cognitive scores that were, on average, 3.29 points higher than those who received the standard formula, suggesting that MFGM can help enhance their cognitive abilities. Additionally, MFGM supplementation showed notable benefits for executive function, which includes skills such as problem-solving and self-control. However, no significant differences between MFGM supplementation and standard infant formula were found in terms of language, motor, and social–emotional development. Our noninferiority analysis revealed that MFGM-supplemented formula is comparable to breast milk in supporting cognitive development. The results suggest that MFGM-supplemented formula can be a valuable alternative for infants who cannot be breastfed.

Previous meta-analyses have shown mixed results regarding other supplements. For instance, a systematic review and meta-analysis by Verfuerden et al. examined the effect of LCPUFA supplementation of infant formula on long-term cognitive function. The study found no significant cognitive benefits of LCPUFA supplementation in either term or preterm-born children [29]. Although the EU Commission recently mandated the addition of docosahexaenoic acid, a type of LCPUFA, to all infant and follow-on formulas, this decision was based on theoretical arguments rather than on solid evidence of cognitive benefits [30]. This supplementation comes at a cost, with families potentially spending up to USD 400 extra per year on LCPUFA-supplemented formulas compared to those that are not supplemented, and mandatory supplementation could result in market-wide price increases [31]. In contrast, our study highlights the striking cognitive benefits of MFGM supplementation, suggesting that MFGM should receive more attention in infant nutrition strategies given its demonstrated positive impact on cognitive development.

The significant improvements in cognitive and executive function development can be attributed to the bioactive components of MFGM, such as sphingomyelin and gangliosides, which are critical for myelination and neural signaling [13,14,32]. Myelination enhances the speed and efficiency of neural transmission, which is crucial for cognitive processes and executive function [13]. These components support automaticity and fluency, which are foundational for complex cognitive tasks, rather than basic accuracy measures more associated with motor or language development [33].

The non-inferiority of MFGM-supplemented formula to breast milk in cognitive development marks a significant advancement in infant nutrition. This finding encourages a shift toward recommending infant formulas that are comparable to breast milk in terms of cognitive benefits [34]. For mothers who cannot breastfeed, the option of such a formula can provide reassurance and meet a critical aspect of their infant’s nutritional needs [35].

Despite the significant cognitive benefits observed, our findings indicated no substantial impact of MFGM supplementation on motor, language, and social-emotional development. This lack of effect may be due to the specific bioactive components of MFGM, which primarily support cognitive and executive functions through myelination and neural signaling rather than influencing the broader spectrum of developmental domains. These results align with previous studies, such as the meta-analysis by Verfuerden et al., which also reported limited effects of other supplements like LCPUFA on various developmental outcomes [29]. Furthermore, the potential benefits for preterm infants should be considered, as they are at higher risk for neurodevelopmental delays. Although our study focused on term infants, the bioactive components in MFGM, which enhance neural development, could potentially offer significant benefits for preterm infants as well. Future research should explore this possibility to provide more comprehensive insights into the role of MFGM in supporting the development of preterm infants.

It is noteworthy that the varied results in our included studies could be influenced by other supplements in the formulas. For example, the Chen (2024) study showed a negative value for cognition [21], while other studies like Colombo [22], Gurnida [23], and Nieto-Ruiz [26] demonstrated positive effects on language development. Similarly, Gurnida [23] and Li [24] reported positive effects on motor development. These differences may be attributed to additional components such as LCPUFAs, lactoferrin, and symbiotics present in the formulas. These additives, known for their potential developmental benefits, could have contributed to the varied outcomes observed across studies. Due to the limited number of studies, we could not analyze these differences in the current study, but this represents an important next step for future research. Understanding the specific impact of each supplement is crucial for optimizing formula compositions to target comprehensive developmental benefits.

Our study has some limitations. The included studies varied in their interventions, follow-up periods, and specific formulations of the MFGM supplements, affecting the generalizability of the results. The lack of significant findings in language, motor, and social-emotional development could be due to differences in measurement tools and study designs, including the timing and type of assessments used. Small sample sizes in some studies limited the power to detect differences, particularly in subgroup analyses. Additionally, the heterogeneity in study populations, such as differences in socioeconomic status, maternal education, and baseline nutritional status, may have contributed to variability in outcomes. Potential publication bias and the lack of control for all confounding factors, such as parental IQ and home environment, also pose limitations. Lastly, dual feeding practices observed in some studies, where infants received both formula and breast milk, introduce another layer of complexity. 

## 5. Conclusions

Our findings reveal that MFGM-supplemented infant formula is significantly associated with enhanced cognitive development and demonstrates noninferiority to breast milk. While further research is needed to confirm its effects on other developmental domains, these findings support MFGM as a valuable addition to infant formula, addressing one of the primary concerns of mothers—the cognitive development of their infants. This advancement marks a new era in infant feeding, offering a viable option for optimizing neurodevelopmental outcomes in formula-fed neonates and infants. Future studies should use larger sample sizes and focus on the long-term outcomes of MFGM supplementation and standardized assessment tools to confirm these findings. Investigating the specific components of MFGM that contribute most effectively to neurodevelopment could further optimize the composition of infant formulas. 

## Figures and Tables

**Figure 1 nutrients-16-02374-f001:**
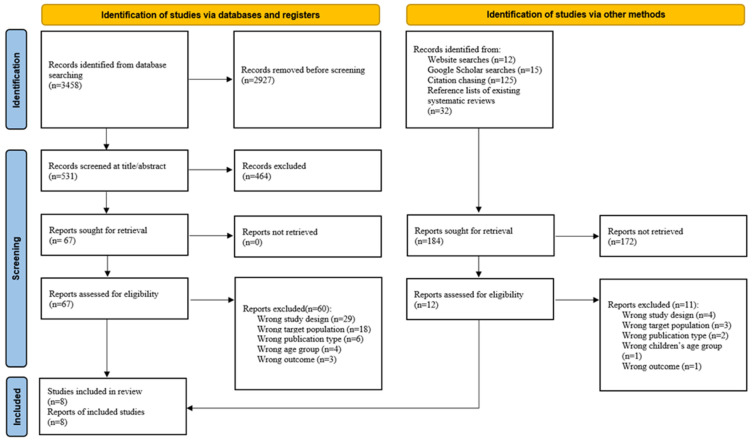
PRISMA 2020 flow diagram.

**Figure 2 nutrients-16-02374-f002:**
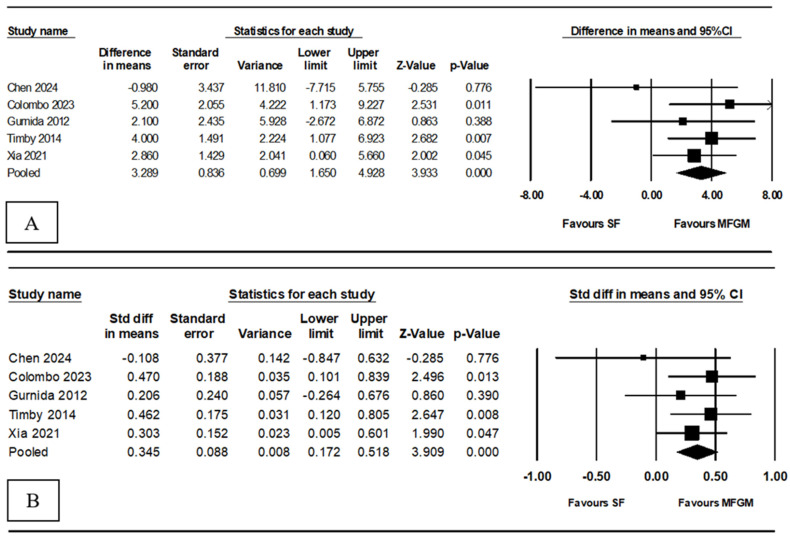
Forest plot of association between MFGM supplementation and cognitive development; (**A**) shows effect size in difference in mean and (**B**) shows effect size in standard mean difference; MFGM, milk fat globule membrane; SF, standard formula [21,22,23,27,28].

**Figure 3 nutrients-16-02374-f003:**
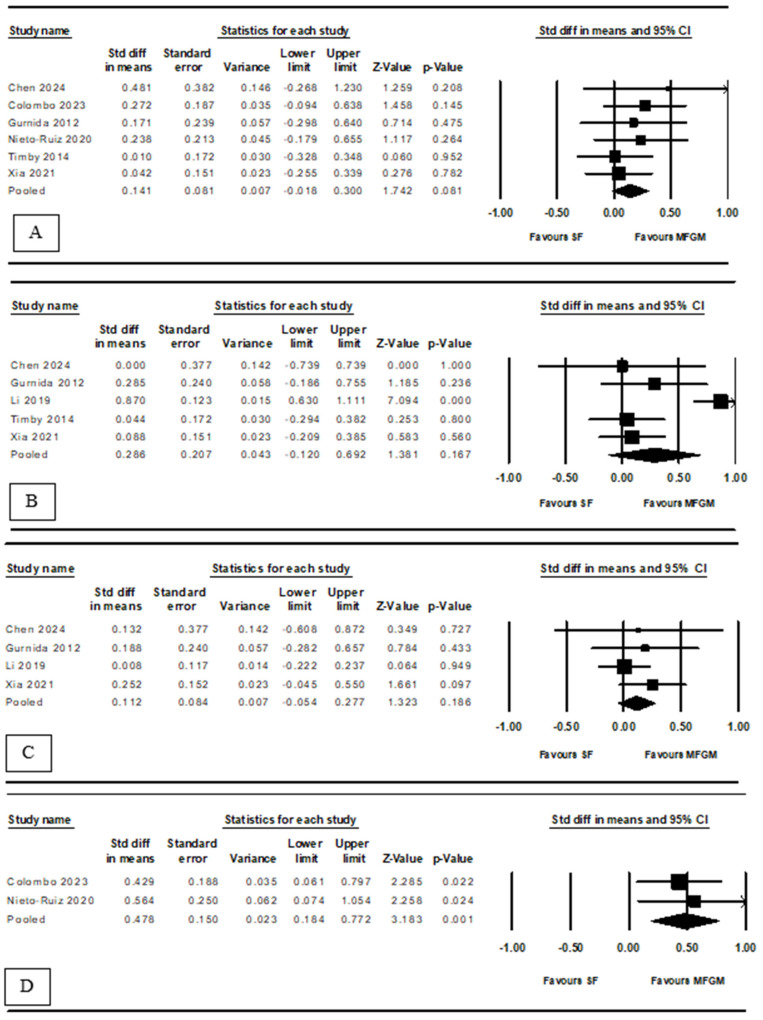
Forest plot of the association between MFGM supplementation and (**A**) language (including data from Nieto-Ruiz et al. 2020 [25]), (**B**) motor, (**C**) social–emotional, and (**D**) executive function development (including data from Nieto-Ruiz et al. 2020 [26]); MFGM, milk fat globule membrane; SF, standard formula [21,22,23,24,25,26,27,28].

**Figure 4 nutrients-16-02374-f004:**
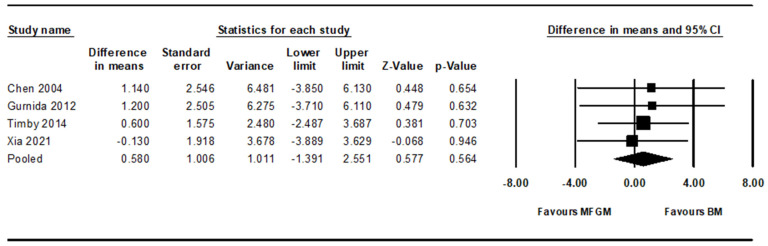
Forest plot of association between MFGM-supplemented infant formula vs. breast milk and cognitive development; MFGM, milk fat globule membrane [21,23,27,28].

**Figure 5 nutrients-16-02374-f005:**
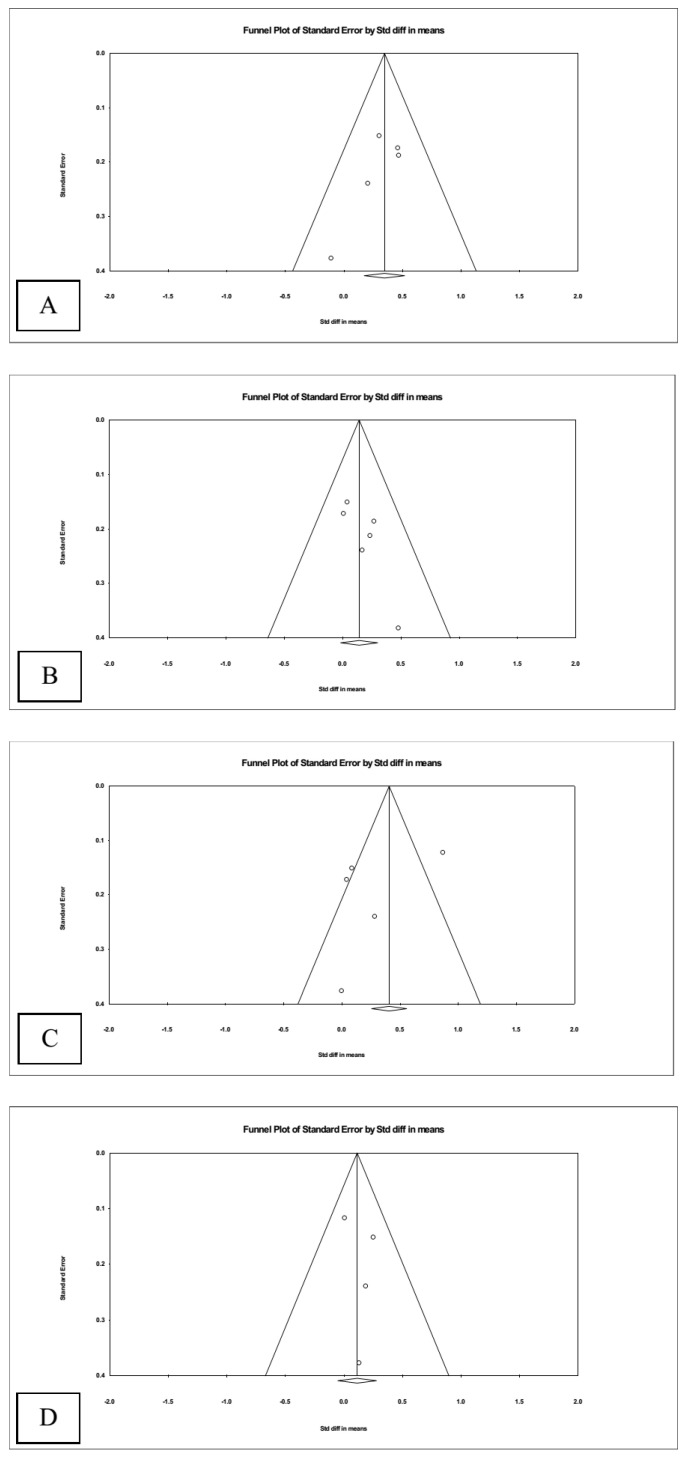
Funnel plots of studies assessing (**A**) cognitive, (**B**) language, (**C**) motor, and (**D**) social–emotional development.

**Figure 6 nutrients-16-02374-f006:**
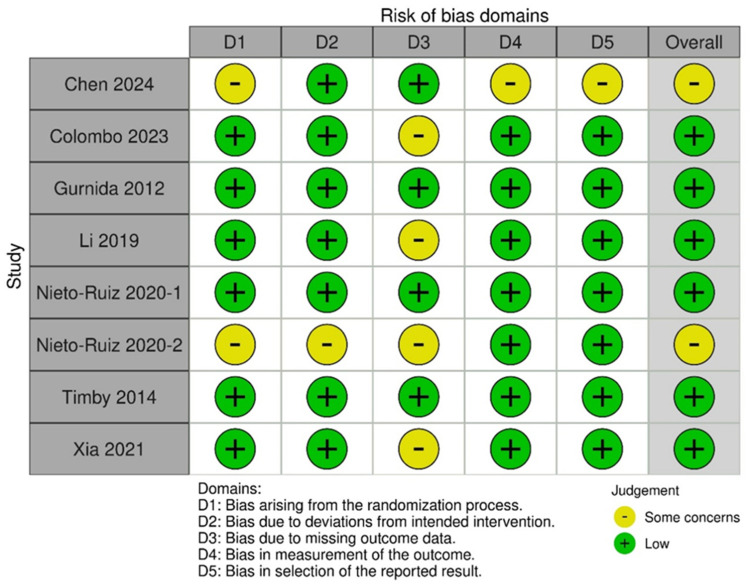
Risk-of-bias plot of each study [21,22,23,24,25,26,27,28].

**Table 1 nutrients-16-02374-t001:** Summary of Included Studies.

Study	Country	N	Population	Intervention	Comparison	Outcome
Chen et al., 2024 [21]	China	79 MFGM = 17 SF = 12 BF = 50	Healthy, term infants	Enriched formula milk powder containing 1,3-dioleoyl-2-palmitoylglycerol and MFGM	SF, BF	At 4 and 6 months old CNBS-R2016
Colombo et al., 2023 [22]	China	116 MFGM = 57 SF = 59	Healthy, term neonates and infants	MFGM + lactoferrin From 10–14 days of age until 365 days of age	SF	At 5.5 years old WPPSI-IV, CBCL, Stroop Task, DCCS
Gurnida et al., 2012 [23]	Indonesia	110 MFGM = 35 SF = 35 BF = 40	Healthy, term neonates and infants	MFGM from enrollment (2–8 weeks of age) up to 6 months (24 weeks of age)	SF, BF	At 6 months old GMDS
Li et al., 2019 [24]	China	191 MFGM = 143 SF = 148	Healthy, term neonates and infants	MFGM + lactoferrin From 10–14 days of age up to 365 days of age	SF	At day 365 and 545 BSID
Nieto-Ruiz et al., 2020-1 [25]	Spain	103 MFGM = 41 SF = 29 BF = 33	Healthy 0–2-month-old full-term neonates, infants, and children	LCPUFAs AA, DHA, MFGM, symbiotics, gangliosides, nucleotides, and sialic acid From 0–2 months of age up to 18 months of age	SF, BF	At 1.5 and 2.5 years old CBCL
Nieto-Ruiz et al., 2020-2 [26]	Spain	122 MFGM = 43 SF = 46 BF = 33	Healthy 0–2-month-old full-term neonates, infants, and children	LCPUFAs AA, DHA, MFGM, symbiotics, gangliosides, nucleotides, and sialic acid From 0–2 months of age up to 18 months of age	SF, BF	At 4 years old PLON-R
Timby et al., 2014 [27]	Sweden	213 MFGM = 73 SF = 68 BF = 72	Healthy neonates and infants	MFGM from enrollment < 2 months to 6 months of age, followed up until 12 months of age	SF, BF	At 12 months old BSID
Xia et al., 2021 [28]	China	418 MFGM = 108 SF = 104 BF = 206	Healthy neonates and infants	MFGM from enrollment (<14 days of age) until 6 months of age	SF, BF	At 6 and 12 months old BSID

Abbreviations: MFGM, milk fat globule membrane; CNBS-R2016, Children Neuropsychological and Behavior Scale-Revision 2016; WPPSI-IV, Wechsler Preschool and Primary Scale of Intelligence—Fourth Edition; CBCL, Child Behavior Checklist; DCCS, Denver Developmental Screening Test; GMDS, Griffiths Mental Development Scales; BSID, Bayley Scales of Infant Development; LCPUFAs, long-chain polyunsaturated fatty acids; AA, arachidonic acid; DHA, docosahexaenoic acid; PLON-R, Preschool Language Scale, Revised; SF, standard formula; BF, breastfeeding.

**Table 2 nutrients-16-02374-t002:** Noninferiority of Infant Formula Supplemented with MFGM Compared to Breast Milk.

Child Development Domain	Mean Difference	95% Lower Bound	95% Upper Bound	Noninferior
Cognitive	0.58	–1.39	2.55	Yes
Language	1.91	–2.046	4.074	Inconclusive
Motor	1.20	–0.66	3.06	Inconclusive
Social–emotional	3.41	–4.65	11.47	Inconclusive

## Data Availability

Therdpong Thongseiratch had full access to all the data in the study and took responsibility for the integrity of the data and the accuracy of the data analysis.

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
