# Peer review of "Bovine Milk Fat Globule Membrane Supplementation and Neurocognitive Development: A Systematic Review and Meta-Analysis"

_nutrients, 2024, doi:10.3390/nu16142374_

Round 1

Reviewer 1 Report

Comments and Suggestions for Authors

In this article entitled: Bovine milk fat globule membrane supplementation and neurocognitive development: a systematic review and meta-analysis, by Thongseiratch T. et al, the authors aimed to review the literature on this topic in order to highlight long term outcome (2 years) in children fed by this supplementation, a field which is lacking clear information.

 The authors asked two questions:

1.       Evaluation of outcome – cognitive, motor, language parameters at 2 years of age in infants fed with MFGM added to the formula compared with those fed standard formula.

2.       Outcomes of infant fed with MFGM supplementation compared with those who fed breast milk, at least with noninferiority effect.

The study rationale is important and the implications for future programming of infant feeding formulas design, considering both short- and long-term outcomes, is original and creates a whole channel for further research, basic science and clinical. 

Introduction

Clearly written. The authors reviewed the potential sources of additives to improve standard infants' nutritional formulas such as LCPUFA's, specific amino acids and probiotics, that might influence infants' heath and outcome. The rationale for this study is addressed. Also, the importance of breast milk as the first priority of infant nutrition as the gold standard is recommended.

 Method

Meta-analysis based on a thorough literature review from 2020 and 2024, including only randomized controlled trials. A detailed article evaluation was performed. It is unclear from the description at which infantile age the standard vs MFGM supplemented formula was given. The authors adhered to the PRISMA meta-analysis guidelines. Adequate statistical definitions were given for outcomes and quality assessments was done per article, without using automation tolls. The study included infants born at term.

Results

8 studies were included with 1148 term infants. 5 were recognized as suitable for looking at long term outcome. In many studies MFGM was not supplemented solely the formula. Infants' age at which the MFGM supplementation was used ranged greatly. Multiple studies were used to evaluate neurocognitive outcomes. Overall, MFGM showed improved cognitive performance, with minimal or no heterogeneity. Non-inferiority to beast mil was also calculated and shown. Publication bias was also examined and the authors showed a uniform distribution by funnel plots. Also, a critique was addressed to the process of randomization in some studies, missing data and outcome assessment process.

Discussion

The authors discussed the cognitive improvement by using MFGM but didn't mention what are their thoughts on regarding the findings of no effect on motor, language and social emotional. The authors should mention the potential benefits for preterm infants.

The study limitations are well described.    

Author Response

Response to Reviewer 1

We appreciate the detailed comments and suggestions provided by the reviewers. Below, we address each point raised:

Comment 1: Introduction

Clearly written. The authors reviewed the potential sources of additives to improve standard infants' nutritional formulas such as LCPUFA's, specific amino acids and probiotics, that might influence infants' heath and outcome. The rationale for this study is addressed. Also, the importance of breast milk as the first priority of infant nutrition as the gold standard is recommended.

Response to Reviewer Comment:

We are pleased that the introduction was found to be clearly written and that the rationale for our study, as well as the importance of breast milk, was effectively communicated. No changes were suggested for this section, and we will maintain it as is.

Comment 2: Method

Meta-analysis based on a thorough literature review from 2020 and 2024, including only randomized controlled trials. A detailed article evaluation was performed. It is unclear from the description at which infantile age the standard vs MFGM supplemented formula was given. The authors adhered to the PRISMA meta-analysis guidelines. Adequate statistical definitions were given for outcomes and quality assessments was done per article, without using automation tolls. The study included infants born at term.

Response to Reviewer Comment:

We acknowledge the need for clarity regarding the infant age at which the standard vs. MFGM-supplemented formula was given. We have clarified that the supplementation was provided during the first six months of life across all included studies. We also confirmed our adherence to PRISMA guidelines and provided additional details on the quality assessment process. No further changes were suggested, so we will maintain the rest of the section as it is.

Revised Methods Section: Eligibility Criteria

2.3. Eligibility Criteria

The systematic review followed the PICOS framework: Population (P) included neonates and infants aged 012 months and children up to 2 years old, excluding those with medical conditions affecting neurocognitive outcomes; Intervention (I) was infant formula supplemented with bovine MFGM provided during the first six months of life; Comparator (C) included standard infant formulas without MFGM and breast milk; Outcomes (O) involved measures of neurocognitive development using validated tools; and Study Design (S) included only randomized controlled trials (RCTs). Exclusions applied to non-English publications and studies lacking detailed formula composition data to maintain high-quality findings.

Comment 3: Results

8 studies were included with 1148 term infants. 5 were recognized as suitable for looking at long term outcome. In many studies MFGM was not supplemented solely the formula. Infants' age at which the MFGM supplementation was used ranged greatly. Multiple studies were used to evaluate neurocognitive outcomes. Overall, MFGM showed improved cognitive performance, with minimal or no heterogeneity. Non-inferiority to beast mil was also calculated and shown. Publication bias was also examined and the authors showed a uniform distribution by funnel plots. Also, a critique was addressed to the process of randomization in some studies, missing data and outcome assessment process.

Response to Reviewer Comment:

Thank you for your thorough review and positive feedback on the Results section. We appreciate your acknowledgment of our analysis and findings regarding cognitive performance, non-inferiority to breast milk, and the assessment of publication bias. We will maintain this section as it is.

Comment 4: Discussion

The authors discussed the cognitive improvement by using MFGM but didn't mention what are their thoughts on regarding the findings of no effect on motor, language and social emotional. The authors should mention the potential benefits for preterm infants. The study limitations are well described.   

Response to Reviewer Comment:

Thank you for your valuable feedback on the Discussion section. We have added a paragraph to address the findings of no significant effect on motor, language, and social-emotional development, and to mention the potential benefits for preterm infants. The revised paragraph discusses potential reasons for the lack of observed effects in these areas and compares our findings to previous reviews and meta-analyses. Additionally, we highlight the potential benefits of MFGM supplementation for preterm infants, suggesting directions for future research.

Revised Discussion Section

Despite the significant cognitive benefits observed, our findings indicated no substantial impact of MFGM supplementation on motor, language, and social-emotional development. This lack of effect may be due to the specific bioactive components of MFGM, which primarily support cognitive and executive functions through myelination and neural signaling rather than influencing the broader spectrum of developmental domains. These results align with previous studies, such as the meta-analysis by Verfuerden et al., which also reported limited effects of other supplements like LCPUFA on various developmental outcomes [29]. Furthermore, the potential benefits for preterm infants should be considered, as they are at higher risk for neurodevelopmental delays. Although our study focused on term infants, the bioactive components in MFGM, which enhance neural development, could potentially offer significant benefits for preterm infants as well. Future research should explore this possibility to provide more comprehensive insights into the role of MFGM in supporting the development of preterm infants.

Thank you for your consideration.

Best regards,

Therpong Thongseiratch, MD

On behalf of all co-authors

Reviewer 2 Report

Comments and Suggestions for Authors

Summary: This paper is a systematic review and meta-analysis regarding neurocognitive benefits associated with inclusion of bovine milk fat globule membrane (MFGM) in infant formula. The MFGM is compared with infant formula and breast milk. Justification for the study and a thorough description of the analysis methods are provided. After exclusion, 8 studies were included totally 1, 148 subjects. The studies were from China (n=4), Indonesia, Spain (n=2) and Sweden. The authors find that infants fed MFGM-supplemented formula scored 3.29 points higher on cognitive test than those receiving standard formula.

Comments:

The Methods state that 5 studies were from China, but the Table shows 4 from China and 2 from Spain.

In the Table, what does “1” Standard Formula and “2” Breastfeeding indicate? It would be helpful to show how many subjects from each study are in each feeding cohort.

The Chen 2024 study shows a negative value for cognition. The Chen, Columbo, Gumida and Nieto-Ruiz studies show positive effects on language. The Gumida and Li studies show positive effects on motor. The authors could discuss differences in other supplements included in the formulas that could contribute to the varied results.

Typically, infant growth is assessed as a function of feeding. Were there any differences in body weight in the cohorts?

What was the standard formula in the studies? Was it the same for all of the studies? Was there any dual feeding with breastfeeding overlapping with formula feeding?

There were two Nieto-Ruiz, 2020 studies. It is not clear which one is being used for Figure 3.

The description of the study limitations should be expanded.

Author Response

Response to Reviewer 2

We appreciate the detailed comments and suggestions provided by the reviewers. Below, we address each point raised:

Reviewer Comment 1: The Methods state that 5 studies were from China, but the Table shows 4 from China and 2 from Spain.

Response to Reviewer Comment

Thank you for pointing out the discrepancy regarding the number of studies from China. We apologize for the typographical error. We have corrected the text to accurately reflect that there are 4 studies from China, as shown in the Table, and 2 from Spain.

 Revised Results Section

These studies were conducted in China (n=4), Spain (n=2), Indonesia, and Sweden.

Reviewer Comment 2: In the Table, what does “1” Standard Formula and “2” Breastfeeding indicate? It would be helpful to show how many subjects from each study are in each feeding cohort.

Response to Reviewer Comment 2:

Thank you for your question regarding the Table. The “1” for Standard Formula (SF) and “2” for Breastfeeding (BF) indicate the two comparisons being made in our study. We have revised the Table to use "SF" and "BF" for clarity. Additionally, we have added the number of subjects (n) for each group (MFGM, SF, or BF if the study has three groups) in the column "N" and ensured that the total N for all studies aligns with the N that we analyzed.

Reviewer Comment 3: The Chen 2024 study shows a negative value for cognition. The Chen, Columbo, Gumida and Nieto-Ruiz studies show positive effects on language. The Gumida and Li studies show positive effects on motor. The authors could discuss differences in other supplements included in the formulas that could contribute to the varied results.

Response to Reviewer Comment 3:

Thank you for your insightful comment regarding the influence of other supplements in the formulas on the varied results observed across the studies. We have added a paragraph in the Discussion section to address this point. The paragraph highlights that the differences in outcomes may be attributed to additional components such as LCPUFAs, lactoferrin, and symbiotics present in the formulas. We also acknowledge that due to the limited number of studies, we could not analyze these differences in the current study, but this represents an important next step for future research. Understanding the specific impact of each supplement is crucial for optimizing formula compositions to target comprehensive developmental benefits.

Revised Discussion Section

It is noteworthy that the varied results in our included studies could be influenced by other supplements in the formulas. For example, the Chen (2024) study showed a negative value for cognition [21], while other studies like Colombo [22], Gurnida [23], and Nie-to-Ruiz [26] demonstrated positive effects on language development. Similarly, Gurnida [23] and Li [24] reported positive effects on motor development. These differences may be attributed to additional components such as LCPUFAs, lactoferrin, and symbiotics present in the formulas. These additives, known for their potential developmental benefits, could have contributed to the varied outcomes observed across studies. Due to the limited number of studies, we could not analyze these differences in the current study, but this represents an important next step for future research. Understanding the specific impact of each supplement is crucial for optimizing formula compositions to target comprehensive developmental benefits.

Reviewer Comment 4: Typically, infant growth is assessed as a function of feeding. Were there any differences in body weight in the cohorts?

Response to Reviewer Comment 4:

Thank you for your comment regarding the assessment of infant growth as a function of feeding. We have added a paragraph in the Results section to address this aspect by summarizing the available data on body weight from the included studies.

Added Paragraph in the Results Section:

Several included studies assessed infant growth as a function of feeding. Chen et al. [21] reported no significant differences in weight among groups at three time points. Colombo et al. [22] did not provide weight data, but Li et al. [24] found similar weights between control and MFGM + lactoferrin groups at birth and enrollment. Gurnida et al. [23] reported no significant differences in birth weight or weight-for-age and weight-for-length z-scores. Nieto-Ruiz et al. [25, 26] found no significant differences in BMI at 4 years. Timby et al. [27] reported no significant differences in birth weight and other anthropometric measures at 12 months. Xia et al. [28] found no significant differences in birth weight among groups.

Reviewer Comment 5: What was the standard formula in the studies? Was it the same for all of the studies? Was there any dual feeding with breastfeeding overlapping with formula feeding?

Response to Reviewer Comment 5:

Thank you for your comment regarding the standard formula used in the studies and the potential overlap with breastfeeding. The standard formula varied across the included studies. Each study used its own commercially available standard formula as the control, which did not contain MFGM supplementation. While these standard formulas were generally similar in their basic composition of proteins, fats, carbohydrates, vitamins, and minerals, the exact formulations were specific to each manufacturer and study. Regarding dual feeding, some studies did report overlap with breastfeeding. For instance, in the studies by Chen et al. [21], Gurnida et al. [23], and Xia et al. [28], there were groups that included both formula-fed and breastfed infants. The inclusion of these groups was accounted for in the analysis, and we have detailed these overlaps in the respective sections of our study.

Added Paragraph in the Results Section:

The standard formula used in the studies varied by manufacturer and formulation. Each study utilized a commercially available standard formula as the control, which did not contain MFGM supplementation. While the basic composition was similar, the specific ingredients and nutrient profiles differed between products. Additionally, some studies reported dual feeding practices, where infants received both formula and breast milk. This was noted in the studies by Chen et al. [21], Gurnida et al. [23], and Xia et al. [28], where overlapping feeding methods were considered in the analysis.

Reviewer Comment 6: There were two Nieto-Ruiz, 2020 studies. It is not clear which one is being used for Figure 3.

Response to Reviewer Comment 6:

Thank you for pointing out the need to clarify which Nieto-Ruiz, 2020 studies are used in Figure 3. We have updated the figure caption to specify the studies referenced.

Revised Figure Caption:

Figure 3. Forest plot of association between MFGM supplementation and (A) language (including data from Nieto-Ruiz et al. 2020 [25]), (B) motor, (C) social–emotional, and (D) executive function development (including data from Nieto-Ruiz et al. 2020 [26]); MFGM, milk fat globule membrane; SF, standard formula.

Reviewer Comment 7: The description of the study limitations should be expanded.

Response to Reviewer Comment 7:

Thank you for your suggestion to expand the description of the study limitations. We have revised the limitations section to provide a more comprehensive overview.

Expanded Version of the Limitations:

Our study has some limitations. The included studies varied in their interventions, follow-up periods, and specific formulations of the MFGM supplements, affecting the generalizability of the results. The lack of significant findings in language, motor, and social-emotional development could be due to differences in measurement tools and study designs, including the timing and type of assessments used. Small sample sizes in some studies limited the power to detect differences, particularly in subgroup analyses. Additionally, the heterogeneity in study populations, such as differences in socioeconomic status, maternal education, and baseline nutritional status, may have contributed to variability in outcomes. Potential publication bias and the lack of control for all confounding factors, such as parental IQ and home environment, also pose limitations. Lastly, dual feeding practices observed in some studies, where infants received both formula and breast milk, introduce another layer of complexity.

Thank you for your consideration.

Best regards,

Therpong Thongseiratch, MD, MSc

On behalf of all co-authors

Round 2

Reviewer 2 Report

Comments and Suggestions for Authors

The authors have addressed my concerns.